# The variable prevalence of bovine tuberculosis among dairy herds in Central Ethiopia provides opportunities for targeted intervention

Gizat Almaw [1,2]*, Andrew J. K. Conlan[3], Gobena Ameni[4,5], Balako Gumi[4], Alemseged Alemu[1], Sintayehu Guta[1], Solomon Gebre[1], Abebe Olani[1], Abebe Garoma[1], Dereje Shegu[1], Letebrhan Yimesgen[1], Demeke Nigussie[6], James L. N. Wood[3], Tamrat Abebe[2], Adane Mihret[2,7], Stefan Berg[8], the ETHICOBOTS consortium¶

1 National Animal Health Diagnostic and Investigation Center, Sebeta, Ethiopia, 2 Department of Microbiology, Immunology and Parasitology, College of Health Sciences, Addis Ababa University, Addis Ababa, Ethiopia, 3 Disease Dynamics Unit, Department of Veterinary Medicine, University of Cambridge, Cambridge, United Kingdom, 4 Aklilu Lemma Institute of Pathobiology, Addis Ababa University, Addis Ababa, Ethiopia, 5 Department of Veterinary Medicine, College of Food and Agriculture, United Arab Emirates University, Al Ain, United Arab Emirates, 6 Ethiopian Institute of Agricultural Research, Addis Ababa, Ethiopia, 7 Armauer Hansen Research Institute, Addis Ababa, Ethiopia, 8 Bacteriology Department, Animal and Plant Health Agency, Weybridge, United Kingdom

¶ Members of the ETHICOBOTS consortium are listed in the Acknowledgments. Consortium lead author: James LN Wood, email: jlnw2@cam.ac.uk
* gizatalm@yahoo.com

**Data Availability Statement:** All relevant data are within the manuscript and its S1 Dataset, S1 Fig, S1 Questionnaire and S1–S4 Tables.

## Abstract

Bovine tuberculosis (bTB) is an important disease for dairy productivity, as well as having the potential for zoonotic transmission. Previous prevalence studies of bTB in the dairy sector in central Ethiopia have suggested high prevalence, however, they have been limited to relatively small scale surveys, raising concerns about their representativeness. Here we carried out a cross sectional one-stage cluster sampling survey taking the dairy herd as a cluster to estimate the prevalence of bTB in dairy farms in six areas of central Ethiopia. The survey, which to date is by far the largest in the area in terms of the number of dairy farms, study areas and risk factors explored, took place from March 2016 to May 2017. This study combined tuberculin skin testing and the collection of additional herd and animal level data by questionnaire to identify potential risk factors contributing to bTB transmission. We applied the single intradermal cervical comparative tuberculin (SICCT) test using >4mm cut-off for considering an individual animal as positive for bTB; at least one reactor animal was required for a herd to be considered bTB positive. Two hundred ninety-nine dairy herds in the six study areas were randomly selected, from which 5,675 cattle were tested. The overall prevalence of bTB after standardisation for herd-size in the population was 54.4% (95% CI 48.7–60%) at the herd level, and it was 24.5% (95% CI 23.3–25.8) at the individual animal level. A Generalized Linear Mixed Model (GLMM) with herd and area as random effect was used to explore risk factors association with bTB status. We found that herd size, age, bTB history at farm, and breed were significant risk factors for animals to be SICCT positive. Animals from large herds had 8.3 times the odds of being tuberculin reactor (OR: 8.3, p-

**Funding:** This research was financially supported by the Ethiopia Control of Bovine Tuberculosis Strategies (ETHICOBOTS) project funded by the Biotechnology and Biological Sciences Research Council, the Department for International Development, the Economic & Social Research Council, the Medical Research Council, the Natural Environment Research Council and the Defence Science & Technology Laboratory, under the Zoonoses and Emerging Livestock Systems (ZELS) programme, ref: BB/L018977/1. Stefan Berg was also funded by Defra, United Kingdom, ref: TBSE3294. The funders had no role in study design, data collection and analysis, decision to publish, or preparation of the manuscript.

**Competing interests:** The authors have declared that no competing interests exist.

value:0.008) as compared to animals from small herds. The effect of age was strongest for animals 8–10 years of age (the oldest category) having 8.9 times the odds of being tuberculin reactors (OR: 8.9, p-value:<0.001) compared to the youngest category. The other identified significant risk factors were bTB history at farm (OR: 5.2, p-value:0.003) and cattle breed (OR: 2.5, p-value: 0.032). Our study demonstrates a high prevalence of bTB in central Ethiopia but with a large variation in within-herd prevalence between herds, findings that lays an important foundation for the future development of control strategies.

## Introduction

Bovine tuberculosis (bTB) is a chronic disease of cattle primarily caused by *Mycobacterium bovis* (*M. bovis*), which has zoonotic potential and can also infect other domestic and wild animals. The disease is prevalent in most of Africa, parts of Asia and the Americas, and in several European countries. Many industrialised countries have managed to reduce or eliminate bTB in their livestock sectors through test-and-slaughter, however significant pockets of infection remain in wildlife [1]. In Africa the disease is endemic due to a lack of control measures. This has economic implications for the growth of the livestock sector, especially the dairy sector, and poses the risk of zoonotic TB transmission which is exacerbated by the existence of concomitant infections such as HIV/AIDS [2]. In Ethiopia, the demand for milk is expanding rapidly due to increased urbanization and population pressure; Ethiopia is the second most populous country in Africa with an estimated population of 110 million people [3]. Since the introduction of intensive dairy farming in central Ethiopia in the 1950s to provide the Emperor and his establishment with milk, the dairy sector has steadily increased. This increase has accelerated during the last 30 years—trying to meet the demand from increased urbanization and the need to supply milk and milk products to the city dwellers [4]. Although the dairy sector is most developed in central Ethiopia, urban centers across the country have more recently seen an increase in dairy farming. This most developed dairy belt in Ethiopia is expected to be challenged with diseases of intensification such as bTB [5, 6]. This is believed to be associated with mainly two factors: Firstly, a shift from dairy herding with existing local zebu cows to crosses of exotic breeds (mainly Holstein Friesian cows), which produce higher milk yields, have established dairy herds that are likely to be more susceptible to bTB [7, 8]. Secondly, an intensified dairy sector with larger herds has likely increased disease transmission as bTB is thriving in an environment with higher density population. bTB animal prevalence recorded in Ethiopia has ranged from around 3% in smallholder production systems (rearing mainly zebu cattle) up to 48% in intensive dairy productions [5, 7, 9–11] and the national average recently estimated to be ~ 5.8% [12].

Tschopp and colleagues [13] estimated (simulated) the cost of bTB for the urban dairy production in central Ethiopia (Addis Ababa) to have ranged from US$500,000–4.9 million over a period of six years (2005–2011). One target for the Ethiopian government in its 2015–2020 Livestock Master Plan is to transform the dairy sector by increasing the number of crossbred cattle by almost eight times the base-year number [14]. Such expansion comes however with a risk since transmission of infectious diseases, such as bTB, is likely to thrive by intensification [12, 15]. This also raises the concern that bTB may spread to the emerging dairies in the regional towns through trading of high milk yield animals from infected farms in the central regions.

Previous bTB prevalence studies in this part of Ethiopia were surveys of smaller scale (significantly fewer farms or fewer study areas) and conducted over different time periods and

study areas, leading to concerns about representativeness. Accordingly, there is likely to have been over/under representation of dairy farms in past surveys due to lack of either appropriate stratified sampling or standardisation of the results [10]. A comprehensive review of bTB in Ethiopia by Sibhat et al. [12] showed limitations of previous prevalence studies, central Ethiopia included, including the scope of study objectives, methodology used, target population and geographic coverage. Therefore we carried out a large scale systematically stratified survey to assess the current status of bTB prevalence in the established dairy sector in central Ethiopia and to identify contributing risk factors for the spread of the disease to inform the development of potential control strategies.

## Materials and methods

### Study areas

Six study areas were purposefully selected in the urban areas of central Ethiopia, including Addis Ababa city, and Sebeta, Holeta, Sululta, Sendafa and Bishoftu towns (Fig 1). Central Ethiopia, which includes the study areas, was a pioneer for the modern dairy development in

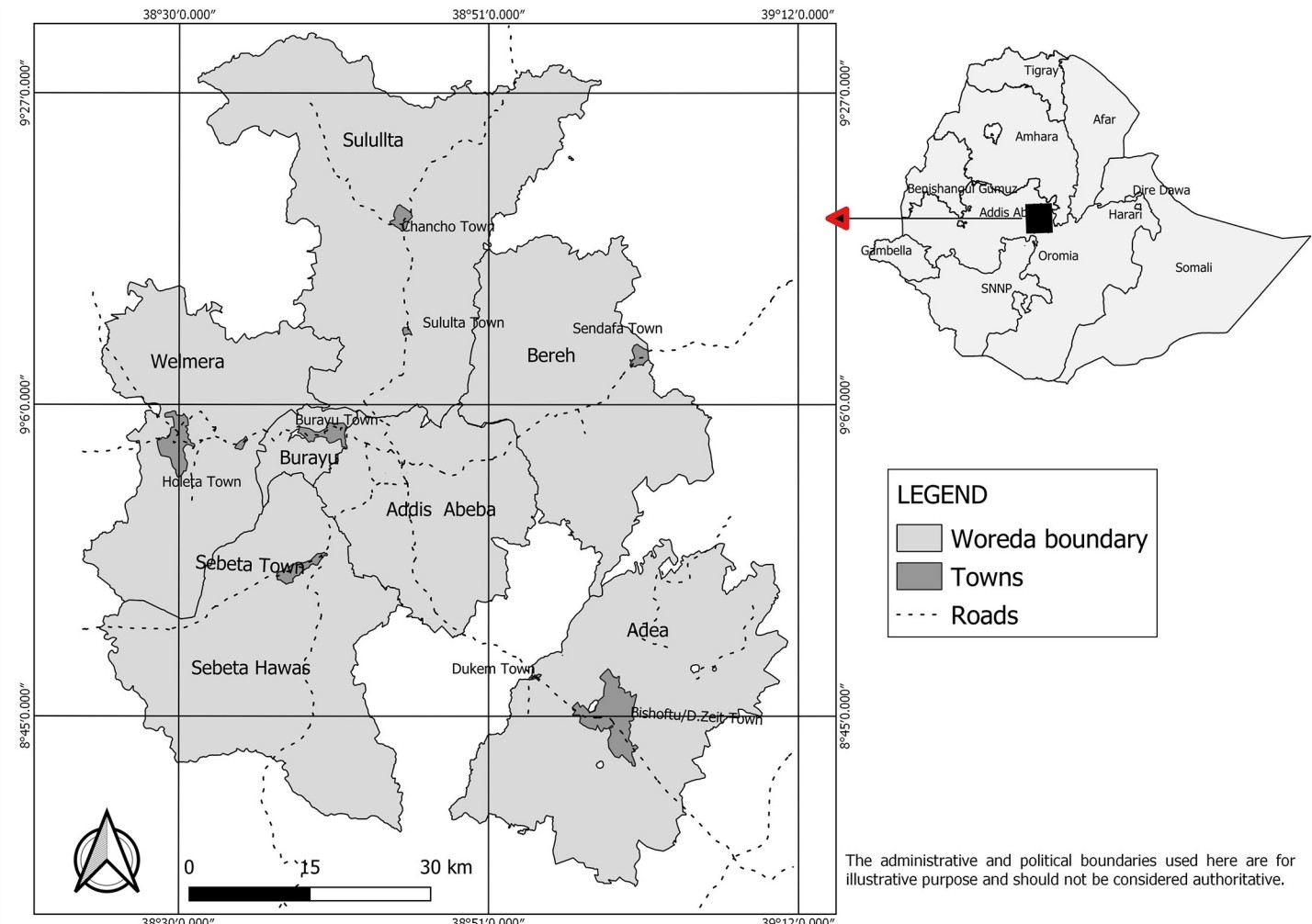

**Fig 1. Map of the study areas: Addis Ababa city and Sululta, Sendafa, Holeta, Sebeta, and Bishoftu towns.**

Ethiopia with the first number of exotic dairy cattle arriving in the early 1950s as a donation from the United Nations [4] and this area has then over decades established itself as the most developed dairy belt in Ethiopia. The study areas are currently the main milk suppliers for people in Addis Ababa and the surrounding peri-urban areas. A free software program called Quantum Geographic Information System (QGIS) version 3.8 [16] was used for compiling the maps. Administrative and road data were extracted and complied from publicly available information of Central Statistical Agency of Ethiopia [17] and Ethiopian Roads Authority [18].

## Study design

This study was a cross sectional study conducted from March 2016 to May 2017. Lists of herds (the sampling frame) were established at the start of the study in collaboration with district veterinary officers in respective study sites. The term "herd" was used to describe the group of cattle that are housed on a holding at the time of data collection [19]. Herds, with the purpose of producing milk and dairy products, having five or more cattle were included and a list of 1,323 herds was established as a sampling frame. The herds were classified as small [5–20], medium [21–37], and large herds [38–168] (168 being the largest herd size in the studied herds) [5].

Inclusion and exclusion criteria for the study herds: Herd size was the criteria used and herds with less than five animals were excluded.

## Sample size and sampling

Sample size was determined using the following formula (that assumes a large population) following one-stage cluster sampling method taking dairy herd as a cluster [20] and every animal in the selected cluster was tested.

$$g = \frac{1.96^2 \{nVC + Pexp)1 - Pexp)\}}{nd^2}$$

Where : g = number of herd to be sampled;

n = predicted average number of animals per herd (n = 13);

Pexp = expected prevalence (Pexp = 0.3 from previous study [5]

d = desired absolute precision (d = 0.05);

VC = between − herd variance (VC = 0.233) [21]

Using the assumed parameter values gives an estimated sample size of 383 farms. This was adjusted down using a small population correction (below) to 298.

$$g_{adj} = \frac{G*g}{G + g}, [20]$$

Where : G = total number of herds

g = the calculated sample size for large herds

$$g_{adj} = \frac{1323*383}{1323 + 383} \approx 298$$

Hence, we tested 299 herds out of 1323 registered herds in the study sites and selection of each herd was random. All animals in the 299 herds (5,675 animals) were tested excluding animals less than 6 weeks of age and pregnant cattle ≥8 months pregnant.

For herd recruitment and sampling of the 299 herds, proportionate sample was obtained using the formula: (sample size/population size) x stratum size (small, medium or large herd)

[20] i.e. 298/1323 = 0.225 x stratum size. In the actual study the fractions for large, medium and small herds were 71% (n = 212), 16% (n = 49) and 13% (n = 38), respectively and those in the overall population were 89%, 7% and 4%. The over-representation of larger farms was due to a greater level of refusal to participate in smaller herds, despite efforts to address this, and numbers were made up in medium and large herds. A direct method of standardization (adjustment) [20] was employed to adjust for the effect of having a higher representation of larger farms in the crude overall bTB prevalence result.

**Single Intradermal Cervical Comparative Tuberculin (SICCT) test.** The procedure of the SICCT test was adapted from OIE Terrestrial Manual, 2009 (Bovine Tuberculosis) and the supplier of Tuberculin PPD was Prionics, Lelystad, The Netherlands. The injection site used was at the border of the anterior and middle thirds of left side (for consistency) of the neck. Two sites were used, one for bovine PPD (lower site) and the other for avian PPD (upper site). The upper site was 10 cm below the crest and the lower site was 12.5 cm from the upper site, on a line drawn parallel with the line of the shoulder. The selected site of injection was shaved to an adequately sized area for identification of the injection sites and cleansed. Before injection, a fold of skin at each of the intended injection sites and within the clipped area was taken between the forefinger and thumb and measured to the nearest millimeter using the same digital caliper (0-150mm range) throughout the survey. Then 0.1 ml of Bovine Tuberculin PPD and 0.1ml of Avian Tuberculin PPD was injected intradermally in the lower and upper site, respectively. A correct injection was confirmed by palpating a small pea-like swelling at each injection site. The two injection sites were re-measured after 72 hours by the same person who measured the skin thickness before the injection. For the interpretation, the SICCT test was considered positive if the difference was more than 4 mm; inconclusive if between 1 to 4 mm; and negative if the increase in skin thickness at the bovine site of injection was less than 1 mm or equal to the increase in the skin reaction at the avian site of injection.

## Farm data collection

Farm data were collected by trained research assistants through face to face interview with pre-tested structured questionnaire to capture animal and herd-level information. General information including herd structure, farm antecedents, farm management/husbandry, housing/ventilation, animal health (veterinary services) and animal bio-security were recorded. Specific information related to potential risk factors for bTB were recorded including animals age, sex, breed, physiology (pregnancy/stages of lactation/body condition), herd size, cattle sourcing (cattle movements in and out of the herd), bTB history on farm, contacts /interactions with neighboring herd/other domestic animals/wild animals etc. (S1 Questionnaire). Global Positioning System(GPS) data was collected for each herd for mapping bTB prevalence in the study areas (S1 Fig).

## Statistical analysis

Data from questionnaires and the tuberculin skin test were curated and coded. All the statistical analysis was performed using the R statistical language [22] and RStudio [23]. Based on the SICCT test, the animal level and herd level bTB prevalence for Addis Ababa city and surrounding five study areas was described and 95% confidence interval calculated. The Kruskal–Wallis test was used for comparison of variability in within herd bTB prevalence (%) among studied dairy herds. Our dataset was hierarchal in nature *i.e.* individual animals were clustered within herds and herds were clustered within study areas. To account for this clustering and deal with variation in prevalence between study areas and in particular between herds, a Generalized Linear Mixed Model (GLMM) [24] was used which allowed us to treat herd and study areas as

random effects with a binary response as an outcome variable (bTB reactor or not reactor). Animals with reading difference between 1–4 mm were treated as negatives. We used the glmer() function in the lme4 package [25]. The statistical unit of analysis was the individual animal. We performed a univariable screen to select variables for inclusion in the multivariable model. All variables with a p-value of < 0.20 and those with a high biological relevance were considered as candidate variables for the model building. These candidate explanatory variables were investigated further for collinearity requiring that all selected variables for the multivariable model have a variance inflation factor (VIF) of < 5 [11]. Statistical significance was set at the 5% level.

For binary data a binomial response (more specifically, the Bernoulli distribution) was used [26]. To specify the model, we define the binary response variable:

$$Y_i = \begin{cases} 1 & \text{If the animal is positive for bTB,} \\ 0 & \text{Otherwise.} \end{cases}$$

$Y_i \text{ Bin } (P_i)$

The probability $P_i$ of the $i^{th}$ animal being bTB positive is :

$$\log\left(\frac{P_i}{1 - P}\right) = \beta_o + \beta X_i + \mu_{herd(i)} + \gamma_{area(i)},$$

Where :

$\beta_o$ is the intercept

$\beta$ is a parameter of fixed effects,

$X$i are explanatory variables values for the $i^{th}$ animal,

$\mu_{herd(i)}$ is the random effect of the herd (which contains animal i),

$\gamma_{area(i)}$ is the random effect of the study area (which contains animal i),

All screened predictors were initially included in the global model, including biologically plausible two-way interactions. Breed was considered as potential confounder for herd size. As some confounding is invariably present, and the important issue is how large the confounding effect is, not whether or not it is present [24]. We specified a difference of 20% change in the odds ratio as an indication of confounding [24]. The removal of breed from the final model changed the logit of herd size by 13.2% (7.7.-6.8)/6.8) for medium herds and by 19.2% (9.9-8.8)/8.3) for large herds, thus no strong confounding effect was found between the two factors.

For model fitting in addition to the global model, a set of models were proposed (S4 Table) to identify potential risk factors that most affect the outcome variable of interest i.e. bTB status. We used the Akaike information criterion (AIC) for comparing and selecting between models. As described by Burnham and Anderson [27], the AIC approach is first to calculate an AIC value for each model proposed and to examine the differences between the AIC values of competing models to the model with minimum value of AIC (often termed as the best model). To put this mathematically: ΔAIC = AIC$i$−minAIC; where AIC$i$ is the competing model and minAIC is the model with the minimum AIC value. We used this ΔAIC value to rank and identify candidate models. A threshold was set for identifying candidate models; where models with ΔAIC < 3 and Akaike weights (w > 0.05) [11] were set as candidate models. A model with highest Akaike weights value (often interpreted as the probability that model is the best model) was used for selecting the best model. In our data we identified that the interaction effect between herd and breed was biasing estimates of other variables (skewing the estimate for the herd size variable) due to the small number of zebu cattle in the medium herd level

category. Dropping this interaction–results in the global model having both the lowest AIC and highest Akaike weight and explained the data well and subsequently selected for reporting.

### Ethical considerations

This study was approved by AHRI-ALERT Ethics Review Committee (Project Reg.No PO46/14) and Ethiopia's National Research Ethics Review Committee (NRERC No. 3.10/800/07). Informed consent was obtained verbally from dairy farm owners who were briefed in the presence of a witness (local experts) on the tuberculin skin testing procedure; no known risks to the animal associated with this; their participation in study is voluntary, and that confidentiality on test result will be maintained. When agreed, the witness and the participant's full addresses including their mobile phone numbers were recorded for filing and in case contact with participant was needed.

## Results

### Description of the herd demography and characteristics

This study investigated 299 dairy herds (212 small, 49 medium, and 38 large farms) for bTB using the SICCT test in the urban and peri-urban areas of central Ethiopia. In addition, descriptive data on these herds were collected. With regard to ownership of the studied herds, 238 (82.9%) herds were owned privately, 31were cooperatives (10.8%), eight were government herds (2.8%) and ten were share companies (3.5%). Twelve herds had no records about ownership. The majority of herds (77.1%) had loose house type and practice zero grazing (roughage with supplement feeding) (78.5%). Artificial insemination (AI) was the main breeding strategy for 69% of these farmers, 83% vaccinated their cattle against major diseases, while 67% dewormed their cattle on a regular basis. The herd structure of the studied dairy herds is presented in Table 1 and additional herds characteristics is provided in S1 Table.

### Prevalence of bTB in the study population

In total 5,675 cattle from 299 herds were tested by using the SICCT test. Overall there were1,776 reactors (31.3% crude animal prevalence- not adjusted for herd size; 95% CI: 30–33) in 180 herds (60.9% crude herd prevalence; 95% CI:55.2–66.2%), with each positive herd having at least one reactor (Table 2). Sebeta had the highest prevalence (42% at animal level with 95% CI: 38–46% and 74%at herd level with 95% CI: 55–87%) among all six regions whereas Holeta had the lowest prevalence(17% at animal level with 95% CI: 14–20% and 27% at herd level with 95% CI: 13–46%). There was significant variation between study areas in prevalence of tuberculin reactors ($\chi^2$ = 143.18, df = 5, p-value <0.001). Using GPS data for individual farms, bTB prevalence maps were created for the six study areas, each visualizing the bTB burden for large, medium and small herds (S1 Fig).

### Herd-size specific prevalence of bTB

The bTB prevalence was stratified on herd size based on the study population (Table 3A). The results showed a different prevalence between herd sizes with a significant increase in prevalence with herd size group. As the recruitment of herds into the study had been somewhat over-represented of larger herds as compared to the original sampling strategy, it was relevant to standardise the prevalence estimates in the study population. Therefore, Table 3B presents herd size specific prevalence of bTB for the standard population (a population we aimed to sample) of all study sites. The overall crude bTB prevalence was higher (31.3%: 95% CI: 30–33)

**Table 1. Herd structure of the 299 studied dairy herds.**

| Characteristics | Levels | Herd size | | | |
|---|---|---|---|---|---|
| | | Small (n = 212) | Medium (n = 49) | Large (n = 38) | Total (n = 299) |
| **Calf (0-1yr)** | Crossbreed | 381 | 257 | 360 | 998 |
| | Zebu | 34 | 5 | 1 | 40 |
| | Exotic (pure) | 0 | 0 | 1 | 1 |
| **Heifer** | Crossbreed | 360 | 191 | 413 | 964 |
| | Zebu | 15 | 4 | 1 | 20 |
| | Exotic(pure) | 0 | 0 | 0 | 0 |
| **Cow** | Crossbreed | 1116 | 703 | 1486 | 3305 |
| | Zebu | 37 | 15 | 52 | 104 |
| | Exotic(pure) | 8 | 2 | 0 | 10 |
| **Bullock/Steers (1–2 yrs)** | Crossbreed | 17 | 27 | 24 | 68 |
| | Zebu | 6 | 2 | 3 | 11 |
| | Exotic (pure) | 0 | 2 | 1 | 3 |
| **Bull/Oxen** | Crossbreed | 32 | 14 | 31 | 77 |
| | Zebu | 52 | 11 | 6 | 69 |
| | Exotic(pure) | 0 | 0 | 5 | 5 |
| **Total cattle** | **Total** | **2058** | **1233** | **2384** | **5675** |
| **Other animals** | Sheep | 549 | 310 | 500 | 1359 |
| | Goats | 99 | 56 | 77 | 232 |
| | Equine | 142 | 24 | 34 | 200 |
| | Dogs | 260 | 87 | 58 | 405 |
| | Cats | 167 | 69 | 18 | 254 |
| | Swine | 6 | 45 | 1511 | 1562 |
| | Poultry | 5963 | 6952 | 7541 | 20456 |

compared with herd size adjusted prevalence (24.5%: 95% CI:23.3–25.8) (using direct method of standardization). The same trend was recorded for the herd level bTB prevalence (Table 3).

## Within herd prevalence of bTB

The average within-herd prevalence is heavily skewed by a relatively small proportion of extremely high prevalence herds (illustrated by Fig 2). Within-herd prevalence is multi-modal with the majority of small and medium herds having a prevalence less than the population average. The population mean 31.5% was higher compared to the median 10%, thus indicating a positive skewedness and that a higher proportion of herds (67.9%) had a within herd prevalence less than the population average. Although the average within-herd prevalence does not demonstrate a strong herd-size dependence, there is a marked difference in the distribution

**Table 2. Animal and herd level bTB prevalence for 299 dairy herds in the six study areas.**

| Level | Addis Ababa | Sebeta | Holeta | Sululta | Sendafa | Bishoftu | Total |
|---|---|---|---|---|---|---|---|
| **Animal level: % Prev. (95%CI)** | 32.8(31–35) | 42.2(38–46) | 16.8 (14–20) | 41.9(38–46) | 25.5(22–30) | 25.5 (23–28) | 31.3(30–33) |
| Positives | 797 | 250 | 90 | 257 | 134 | 248 | 1776 |
| Total number tested | 2432 | 593 | 537 | 614 | 525 | 974 | 5675 |
| **Herd level: % Prev.(95%CI)** | 63 (55–70) | 74 (55–87) | 30 (13–46) | 60(39–78) | 54(33–74) | 73.3(50–85) | 60.9(54–66) |
| Positives | 100 | 23 | 9 | 15 | 13 | 22 | 182 |
| Total number tested | 159 | 31 | 30 | 25 | 24 | 30 | 299 |

**Table 3. Prevalence of bTB stratified by herd-size for (A) the study population and (B) the standard population of the study areas.**

| A | | Study population | | | |
|---|---|---|---|---|---|
| | Herd size group | Herds sampled | Population | bTB positives | Prevalence % (95% CI) |
| **Animal Level** | Small herds (>4 to ≤20) | 212 | 2058 | 373 | 18.1 (16.5–19.4) |
| | Medium herds (>20 to ≤37) | 49 | 1233 | 402 | 32.6 (30–35.3) |
| | Large herds (>37 to ≤168) | 38 | 2384 | 1001 | 42.0 (40–43.9) |
| | Total | 299 | 5675 | 1776 | **31.3** (30–33) |
| **Herd Level** | Small herds (>4 to ≤20) | 212 | 212 | 108 | 50.9(44.3–57.6) |
| | Medium herds (>20 to ≤37) | 49 | 49 | 41 | 83.7(71–91.5) |
| | Large herds (>37 to ≤168) | 38 | 38 | 33 | 86.8 (72.7–94.2) |
| | Total | 299 | 299 | 182 | **60.9**(55.2–66.2) |
| B | | Standard population | | | |
| | Herd size group | Expected herds sampled | Expected population [a] | Expected bTB positives [b] | Expected Prevalence % (95% CI) |
| **Animal Level** | Small herds (>4 to ≤20) | 266 | 2926 | 530 | 18.1 (16.8–19.6) |
| | Medium herds (>20 to ≤37) | 21 | 609 | 199 | 32.7 (29.1–32.7) |
| | Large herds (>37 to ≤168) | 11 | 792 | 333 | 42 (38.7–45.5) |
| | Total | 298 | 4327 | 1062 | **24.5** (23.3.-25.8) |
| **Herd Level** | Small herds (>4 to ≤20 | 266 | 266 | 134 | 50.4(44.4–56.3) |
| | Medium herds (>20 to ≤37) | 21 | 21 | 18 | 85.7(65.4–95.0) |
| | Large herds (>37 to ≤168) | 11 | 11 | 10 | 90.9(62.3–98.4) |
| | Total | 298 | 298 | 162 | **54.4**(48.7–60) |

[a]Expected population = Expected herds sampled * Average population size (for each herd size group)

[b]Expected bTB positives = Expected population * Prevalence in study population (for each herd size group)

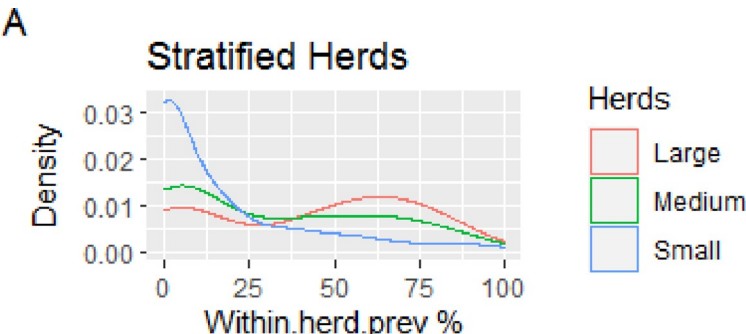

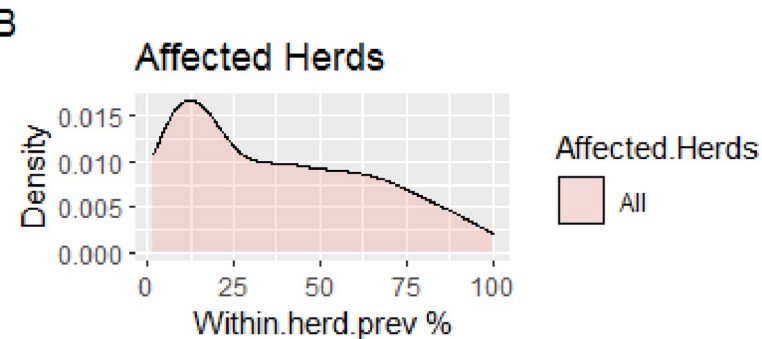

**Fig 2.** (A) Within-herd bTB prevalence distribution for stratified herds (Visualizing multiple distributions simultaneously) (B) Within-herd bTB prevalence distribution for affected herds (bTB prevalence > 0).

with a markedly higher proportion of herds having a prevalence greater than the population average. A greater proportion of large herds (65.8%) (median: 50%) were having within herd prevalence greater than the population average.

Translating this into numbers: the mean within herd prevalence for all herds was 31.5± 30.7% and a median of 10% (lower quartile0 and 42.5% upper quartile). Stratification on large, medium, and small herds, there was a mean within herd prevalence of 40.6%, 35.1%, and 18.8%, respectively, while the median value for the respective herd size was50%, 33%, and 8.3. In this study, there was a significant difference in within herd prevalence among studied dairy herds (Kruskal–Wallis test: df = 2,$\chi^2$ = 33.295, p value < 0.001).

## Risk factor analysis

Sixteen potential risk factors, based on knowledge and understanding of the husbandry system and biological relevance were considered and screened by univariable analysis (Table 4). Twelve variables with p-value of < 0.20 and with OR > 1 were selected for multivariable analysis. Contact with other domestic animals, stages of lactation, viral disease outbreak, and regular de-worming did not fulfil the stated criteria and were excluded from analysis. A full description of the measured risk factors is provided in S2 Table. Total number of examined animals (3[rd] column in Table 4) used for analysis of respective risk factor may differ from the overall number of animals tested (N = 5,675) due to missing values.

**Multivariable analysis of potential risk factors for positive cattle reactors using GLMM with herd and area as random effect.** Based on their high OR, absence of collinearity and statistical significance (p-value <0.2), twelve variables (Table 4) were considered in the final multivariable model. The final model thus consisted of four variables: herd size, age, bTB history at farm, and breed as significant risk factors for bTB. Animals from large herds had 8.3 times the odds of being a bTB tuberculin reactor compared to animals living in small herds. There was also a strong effect of age, with animals 8–10 years of age having 8.9 times the odds of being reactors compared to the youngest category (Table 5).

## Discussion

In this study we set out to perform the largest bTB prevalence study so far in dairy farms in central Ethiopia (Fig 1) to get a comprehensive understanding of the scope of the burden of the disease and identify potential risk factors contributing to the transmission of bTB within the study area. Previous studies had limitations especially in methodology used. For example two studies in Addis Ababa did not show clearly how sample size was determined (no mention of formula and parameters used) and how different herd categories were proportionally sampled [10, 28]. In these studies there was over representation of farms with herd sizes of 20 and above (>25%) where the proportion of these farms in the overall population was estimated to below 10%. There were similar limitations in scope of study objectives, methodology used, target population and geographic coverage as reviewed by Sibhat et al. [12]. Our study therefore addressed the concerns of previous studies. With an overall crude animal prevalence of 31.3% (n = 1,776) (herd size adjusted: 24.5%) and a 60.9% (n = 180) crude prevalence at herd level (herd size adjusted: 54.4%), we recorded a high level of bTB prevalence. However, there was variation between the six study areas: relatively low prevalence was recorded in Holeta and this could be related to earlier work to control for bTB in selected infected government herds in that area, which at the time were supplying heifers to surrounding farmers [29]. In this survey we also noted significant variation of within-herd bTB prevalence (P-value < 0.05) among the studied dairy herds, which ranged from 0 to 100% and with a mean for all herds of 31.5% ± 30.7 SD. This variability would mean differences in transmission due to husbandry and other

**Table 4. Univariable analysis of potential risk factors for cattle tuberculin reactors.**

| Risk factors | Level | Proportion % (bTB positives/total examined) | OR (95% CI) | P value |
|---|---|---|---|---|
| **Herd size** | >4 to ≤20 | 18.1 (373/2058) | ref | |
| | >20 to ≤37 | 32.6 (402/1233) | 2.2 (1.8–2.6) | <0.001 |
| | >37 to ≤168 | 42 (1001/2384) | 3.3 (2.8–3.8) | <0.001 |
| **Age (yrs)** | >0.1 to ≤2 | 21.3 (422/1980) | ref | |
| | >2 to ≤4 | 33.1 (470/ 1420) | 1.8 (1.5–2.1) | <0.001 |
| | >4 to ≤6 | 34.3(376 /1095) | 1.9 (1.6–2.3) | <0.001 |
| | >6 to ≤8 | 39.7(224/564) | 2.4 (1.9–3) | <0.001 |
| | >8 to ≤10 | 41.6 (82/19) | 2.6 (1.9–3.6) | <0.001 |
| **Source** | On farm bred | 30 (1431/4757) | ref | |
| | Purchased | 37.5 (344/916) | 1.4 (1.2–1.6) | <0.001 |
| **Breed** | Zebu | 7.8 (19/244) | ref | |
| | Cross and exotic | 32.3 (1757/5431) | 5.7 (3.6–9.4) | <0.001 |
| **Sex** | Male | 18 (78/433) | ref | |
| | Female | 32.4 (1698/5242) | 2.2 (1.7–2.8) | <0.001 |
| **Farm age (yrs)** | >4to ≤20 | 25.4 (695/2736) | ref | |
| | >20 to ≤35 | 36.6 (715/1951) | 1.7 (1.4–1.9) | <0.001 |
| | >35 to ≤68 | 30 (213/708) | 1.3 (1–1.5) | 0.01 |
| **bTB history at farm** | No | 33.4 (538/1607) | ref | |
| | Yes | 40.8 (381/932) | 1.4 (1.1–1.6) | <0.001 |
| **Contact with other domestic animals** | No | 31.5 (254 /806) | ref | |
| | Yes | 32.5 (702/2161) | 1.04 (0.8–1.2) | 0.64 |
| **Stocking density (no. cattle/m²)** | Less | 28.6 (1314/4601) | ref | |
| | Satisfactory | 35.4 (34/96) | 0.7 (0.5–1.1) | 0.14 |
| | High | 39.8 (300/753) | 1.2 (0.8–1.9) | 0.4 |
| **Ventilation** | Very good | 28.6 (608/2127) | ref | |
| | Satisfactory | 29.7 (506/1706) | 1 (0.9–1.2) | 0.46 |
| | Poor | 34.9 (548/1572) | 1.3 (1.2–1.5) | < 0.001 |
| **Viral disease outbreak** | Yes | 30.6 (851/2784) | ref | |
| | No | 31.2 (867/2728) | 0.9 (0.8–1.1) | 0.35 |
| **Biosecurity measures** | Present | 26.4 (384/1457) | ref | |
| | Absent | 32.8 (1349/4109) | 1.4 (1.1–1.6) | < 0.001 |
| **Neighbor herd** | No | 21.5 (106/494) | ref | |
| | Yes | 31.4 (1527/4857) | 1.7 (1.3–2.1) | <0.001 |
| **House type** | Cubicle | 21.4 (281/1313) | ref | |
| | Loose | 34.5 (1329/3856) | 1.9 (1.6–2.2) | <0.001 |
| | Free movement | 27.2 (94/345) | 1.4 (1–1.8) | 0.02 |
| **Regular de-worming** | No | 35.3 (428/1212) | ref | |
| | Yes | 29.2 (1239/4247) | 0.8 (0.7–9.9) | <0.001 |
| **Stages of lactation (months)** | >0 to ≤2 | 34.7 (137/395) | ref | |
| | >2 to ≤4 | 36.2 (179/494) | 1.1(0.8–1.4) | 0.63 |
| | >4 to ≤8 | 39.2 (304/776) | 1.2(0.6–1.6) | 0.13 |

risk factors discussed in this paper or as reviewed by Broughan et al. [30]. By herd stratifica-
tion, large herds recorded the highest within-herd prevalence (mean: 40.6%) and a larger pro-
portion (65.8%) had a within-herd prevalence greater than the population average. Such high
herd prevalence could be due to an increased risk of within-herd transmission in farms with

**Table 5. GLMM multivariable analysis of potential risk factors for bTB positive cattle using herd and area as random effect.**

| Risk factor | Level | OR (95% CI) | P value |
|---|---|---|---|
| **Herd size** | >4 to ≤20 | ref | |
| | >20 to ≤37 | 6.8 (2.6–17.9) | 0.001 |
| | >37 to ≤168 | 8.3 (2.2–31.5) | 0.008 |
| **Age (yrs)** | >0.1 to ≤2 | ref | |
| | >2 to ≤4 | 2.7.1 (2.1–3.6) | <0.001 |
| | >4 to≤ 6 | 3.5 (2.6–4.8) | <0.001 |
| | >6 to ≤8 | 5 (3.5–7.2) | <0.001 |
| | >8 to ≤10 | 8.9 (5–15.6) | <0.001 |
| **bTB history at farm** | No | ref | |
| | Yes | 5.2 (2.1–12.9) | 0.003 |
| **Breed** | Zebu | ref | |
| | Crossand exotic | 2.5 (1.2–4.5) | 0.032 |

larger herd size [31]. This finding is relevant for control measures such as limited test and removal which could be economically viable in the lower prevalence herds.

Risk factors influence transmission and can be categorised at regional, herd, and animal level [32] and vary across regions for several reasons, such as difference in farm management practices [33]. Analysis of this can be useful to develop a strategy for risk-based surveillance and control for bTB. The present study has identified several risk factors for bTB. Animals from large herds had 8.3 times the odds of being tuberculin reactor compared to those from small herds. Herd size is the most frequently reported risk factor for bTB in Ethiopia and elsewhere [5, 10, 11, 34, 35]. The risk of infection in a herd increases with herd size and this could be due to overcrowding which increases probability of contact between animals in larger herds implying that transmission may be density dependent [30]. High density creates favorable environment for bTB as aerosol is one main route of transmission. The postmortem data collected by Firdessa and colleagues [5] support this as most animals had TB lesions in lungs and/ or lung associated lymph nodes. Also, larger herds often have a larger grazing area, which may expose them to greater environmental risk factors (e.g. wildlife reservoir though not confirmed in Ethiopia) and may also expose them to more neighboring herds [35]. Although the number of large herds in Ethiopia are few (even in the central part of the country) their impact on bTB transmission is likely to be significant as many of them are highly infected and they are primary suppliers of heifers to smallholder farms as well as of milk to consumers and could therefore be most potential sources of infection. If a future bTB control program in Ethiopia would focus on these farms, such intervention could possibly be financially affordable given their small number and turning them into bTB free herds could potentially have a significant impact on the overall bTB prevalence in the Ethiopian dairy sector.

When looking for other potential risk factors, there was also a strong effect of age. Animals between 8–10 years old were having the highest odds of being bTB reactors (OR: 8.9, 95% CI: 5–15.6) compared to the baseline category, which was the youngest age group. A linear increase between bTB infection and age was reviewed by Broughan et al. [30] and observed in slaughterhouse surveillances in cattle in Northern Ireland and Great Britain [26, 36]. The mean age of reactor cattle was 4.4 years (95% CI: 4.29–4.56). Longevity increases probability of exposure and it also increases the chance for development of visible TB lesions and detection in slaughterhouse surveillances. In addition, purchase of older cattle—particularly from high

risk areas—could increase the risk of introducing bTB in a herd. Instead, the adoption of risk-based trading has the potential to reduce the risk of bTB spread [37].

We found also that animals from herds with history of bTB had 5.2 times odds of disease detection compared to herds with no history of bTB. In a tuberculin positive herd which did not remove reactors after skin testing, there could be an increase in infection and hence reactor animals. Even in herds which did cull the reactors, there could be recurrent incidents attributable to persistence of infection in such herds due to failure to detect and remove all infected cattle associated with the performance of the skin test [30].

Exotic and cross bred cattle are known to be more susceptible to bTB [8, 30]. Here we found 2.5 times (95% CI: 1.5–5.8) odds of being bTB reactor in these breeds compared to the indigenous zebu breed. The strategy to meet high milk demand is still geared towards improved dairy cattle as a crossbred dairy cow produces on average at least five times more milk than an indigenous zebu cow [38]. With the Ethiopian Government setting a policy to significantly increase the number of crossbred cattle, intensification is likely to increase and thereby the risk of bTB transmission [12, 15]. The final important risk factor we identified is the introduction of cattle to the herd through purchase. We found that cattle purchased from another farm were more often reactors (37.5%) compared to cattle bred at own farm (30%). Although this difference is not statistically significant, it warrants further investigation.

Overall, when comparing our study with previous surveys of dairy cattle in this established dairy belt of Ethiopia, there was no major difference in bTB animal prevalence but our study showed a slight increase in herd prevalence. Firdessa and colleagues [5] recorded in 2009/2010 a 30% (n = 2,956) animal and 50% (n = 88) herd level bTB prevalence while Tsegaye and colleagues [10] in 2006/2007 recorded 34.1% (n = 1,132) animal and 53.6% (n = 56) herd bTB prevalence, respectively, which is comparable to our corresponding figures. This consistency over time suggests that bTB has reached an endemic equilibrium in these herds. The burden of bTB in the dairy belt in central Ethiopia (31%) is much greater than for emerging dairies in regional states, estimated to range from 0.3% to 12% animal prevalence [6, 11, 13, 34]. At present Ethiopia has no bTB control program but if implemented should consider the central region of the country as a bTB high risk area and this report opens up for a scientific approach for future risk-based surveillance and disease intervention. Cattle trading from this region pose high risk of introducing bTB infection to new herds and underlines the significance of cattle trade regulation with pre-movement testing. The significantly lower bTB prevalence recorded in many emerging dairies in the regional states (which could be considered as low risk regions) presents an opportunity for intervention *e.g.* by trade restrictions to prevent further disease transmission from high risk areas like central Ethiopia and introduce testing to support farmers to keep their herds free from bTB. A recent survey by Mekonnen and colleagues [6] recorded an average disease rate of 5.2% (95% CI: 4–6%) in three emerging dairies in regional states, including Hawassa (3%), Gondar (1.4%), and Mekelle (12%). An earlier report from 2014 [39] documented also lower prevalence (below 7%) in eight out of twelve emerging dairies, but ranging from 0.8% to 24% with a few hot spots in Kombolcha (24%) and Mekelle (14%), the latter confirmed by Mekonnen et al. [6]. The lower bTB rates in many of these emerging dairy regions could be due to less cattle movement from high risk regions and less intensification, as they may have emerged more recently. However, if these emerging dairy regions will intensify, and without a strategy for bTB disease control in Ethiopia, it is likely that these regions will be more affected by bTB in the future.

As the Ethiopian dairy sector is expanding, especially through emerging new dairies around many other urban centers across the country, the findings from this study add useful epidemiological information critical for the application of targeted evidence-based control measures.

Therefore, there is now an opportunity to take steps towards a strategy that can control or significantly reduce the burden of bTB in Ethiopia to improve animal and human health.

As a limitation of this study; in some of the herds, which lacked records-for some risk factors such as age, data was collected through interview. As people may not always recall correct information especially for older animals we tried to compliment such age estimation with parity and dentition data.

## Conclusions

The present study reported a high level of bTB prevalence in the large dairy belt around the capital Addis Ababa in central Ethiopia based on the SICCT test. High variability in burden of infection among the tested dairy herds was also an important finding of this study as it can have impact on future disease intervention strategies. In addition, it identified herd size, animal age, cattle breed, and bTB history at farm as important risk factors contributing to the high prevalence of bTB in the central parts of the country.

## Supporting information

**S1 Fig. Map showing the geographical locations and the sizes of bTB-positive and negative herds and within-herd prevalence in central Ethiopia.**
(TIF)

**S1 Table. Characteristics of studied farms.**
(DOC)

**S2 Table. Description of risk factors.**
(DOC)

**S3 Table. Collinearity test of candidate explanatory variables.**
(DOC)

**S4 Table. Summary of the global and candidate GLMM models (candidate models in bold).**
(DOC)

**S1 Questionnaire. Questionnaire for collection of epidemiological data of bovine tuberculosis in central Ethiopia.**
(DOC)

**S1 Dataset. Raw data of this study.**
(CSV)

## Acknowledgments

We thank NAHDIC for their logistical support. The authors would like to acknowledge Drs Matios Lakew, Nebyou Kassa, Worku Birhanu, Getachew Tuli and Asamnew Tesfaye, Bekele Yalew, Mekedes Tamiru, Daniel Tekeste, Chala Dimma, Teferi Benti, Demessa Negessu, all participating dairy farmers, district veterinary officers and institutions for their support during the field work. The members of the ETHICOBOTS consortium are: Abraham Aseffa, Adane Mihret, Bamlak Tessema, Bizuneh Belachew, Eshcolewyene Fekadu, Fantanesh Melese, Gizachew Gemechu, Hawult Taye, Rea Tschopp, Shewit Haile, Sosina Ayalew, Tsegaye Hailu, all from Armauer Hansen Research Institute, Ethiopia; Rea Tschopp from Swiss Tropical and Public Health Institute, Switzerland; Adam Bekele, Chilot Yirga, Mulualem Ambaw, Tadele

Mamo, Tesfaye Solomon, all from Ethiopian Institute of Agricultural Research, Ethiopia; Tilaye Teklewold from Amhara Regional Agricultural Research Institute, Ethiopia; Solomon Gebre, Getachew Gari, Mesfin Sahle, Abde Aliy, Abebe Olani, Asegedech Sirak, Gizat Almaw, Getnet Mekonnen, Mekdes Tamiru, Sintayehu Guta, all from National Animal Health Diagnostic and Investigation Center, Ethiopia; James Wood (consortium lead author), Andrew Conlan, Alan Clarke, all from Cambridge University, United Kingdom; Henrietta L. Moore and Catherine Hodge, both from University College London, United Kingdom; Constance Smith at University of Manchester, United Kingdom; R. Glyn Hewinson, Stefan Berg, Martin Vordermeier, Javier Nunez-Garcia, allfrom Animal and Plant Health Agency, United Kingdom; Gobena Ameni, Berecha Bayissa, Aboma Zewude, Adane Worku, Lemma Terfassa, Mahlet Chanyalew, Temesgen Mohammed, Miserach Zeleke, all from Addis Ababa University, Ethiopia.

## Author Contributions

**Conceptualization:** Gizat Almaw, Gobena Ameni, James L. N. Wood, Adane Mihret, Stefan Berg.

**Data curation:** Gizat Almaw, Andrew J. K. Conlan, Demeke Nigussie.

**Formal analysis:** Gizat Almaw, Andrew J. K. Conlan, Stefan Berg.

**Funding acquisition:** Gobena Ameni, James L. N. Wood, Adane Mihret, Stefan Berg.

**Investigation:** Gizat Almaw, Alemseged Alemu, Sintayehu Guta, Abebe Olani, Abebe Garoma, Dereje Shegu, Letebrhan Yimesgen.

**Methodology:** Gizat Almaw, Andrew J. K. Conlan, Gobena Ameni, Sintayehu Guta, Abebe Olani, Abebe Garoma, Dereje Shegu, James L. N. Wood, Adane Mihret, Stefan Berg.

**Project administration:** Gobena Ameni, Solomon Gebre, James L. N. Wood, Adane Mihret, Stefan Berg.

**Resources:** Andrew J. K. Conlan, Gobena Ameni, Solomon Gebre, James L. N. Wood, Stefan Berg.

**Supervision:** Gobena Ameni, Balako Gumi, James L. N. Wood, Tamrat Abebe, Adane Mihret, Stefan Berg.

**Validation:** Andrew J. K. Conlan.

**Writing – original draft:** Gizat Almaw.

**Writing – review & editing:** Andrew J. K. Conlan, Gobena Ameni, Balako Gumi, James L. N. Wood, Tamrat Abebe, Adane Mihret, Stefan Berg.

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
