## [Decision Letter · Decision Letter 0]

25 Mar 2021

PONE-D-20-28942

High level of bovine tuberculosis in dairy herds of central Ethiopia: a call for intervention

PLOS ONE

Dear Dr. Almaw,

Thank you for submitting your manuscript to PLOS ONE. After careful consideration, we feel that it has merit but does not fully meet PLOS ONE’s publication criteria as it currently stands. Therefore, we invite you to submit a revised version of the manuscript that addresses the points raised during the review process.

Both expert reviewers have highlighted several concerns that preclude the acceptance of the manuscript as it stands. Please revise the manuscript accordingly making sure to address all reviewers' comments.

We look forward to receiving your revised manuscript.

Kind regards,

Angel Abuelo, DVM, MRes, MSc, PhD, DABVP (Dairy), DECBHM

Academic Editor

PLOS ONE

Journal Requirements:

2. Please provide additional details regarding participant consent. In the ethics statement in the Methods and online submission information, please ensure that you have specified (1) whether consent was informed and (2) what type you obtained (for instance, written or verbal, and if verbal, how it was documented and witnessed). If the need for consent was waived by the ethics committee, please include this information.

"This research was financially supported by the Ethiopia Control of Bovine Tuberculosis

Strategies (ETHICOBOTS) project funded by the Biotechnology and Biological Sciences

Research Council, the Department for International Development, the Economic & Social

Research Council, the Medical Research Council, the Natural Environment Research Council

and the Defence Science &Technology Laboratory, under the Zoonoses and Emerging Livestock

Systems (ZELS) program, ref: BB/L018977/1.Stefan Berg was also funded by Defra, United

Kingdom, ref: TBSE3294"

4. One of the noted authors is a group or consortium [ETHICOBOTS consortium]. In addition to naming the author group and listing the individual authors and affiliations within this group in the acknowledgments section of your manuscript, please also indicate clearly a lead author for this group along with a contact email address.

6. We note that Figure 1 and S1Fig in your submission contain map images which may be copyrighted. All PLOS content is published under the Creative Commons Attribution License (CC BY 4.0), which means that the manuscript, images, and Supporting Information files will be freely available online, and any third party is permitted to access, download, copy, distribute, and use these materials in any way, even commercially, with proper attribution. For these reasons, we cannot publish previously copyrighted maps or satellite images created using proprietary data, such as Google software (Google Maps, Street View, and Earth). For more information, see our copyright guidelines: http://journals.plos.org/plosone/s/licenses-and-copyright.

6.1.    You may seek permission from the original copyright holder of Figure 1 and S1Fig to publish the content specifically under the CC BY 4.0 license. 

6.2.    If you are unable to obtain permission from the original copyright holder to publish these figures under the CC BY 4.0 license or if the copyright holder’s requirements are incompatible with the CC BY 4.0 license, please either i) remove the figure or ii) supply a replacement figure that complies with the CC BY 4.0 license. Please check copyright information on all replacement figures and update the figure caption with source information. If applicable, please specify in the figure caption text when a figure is similar but not identical to the original image and is therefore for illustrative purposes only.

Reviewers' comments:

Reviewer's Responses to Questions

**Comments to the Author**

1. Is the manuscript technically sound, and do the data support the conclusions?

Reviewer #1: Yes

Reviewer #2: Yes

2. Has the statistical analysis been performed appropriately and rigorously? 

Reviewer #1: Yes

Reviewer #2: Yes

3. Have the authors made all data underlying the findings in their manuscript fully available?

Reviewer #1: Yes

Reviewer #2: Yes

4. Is the manuscript presented in an intelligible fashion and written in standard English?

Reviewer #1: Yes

Reviewer #2: Yes

5. Review Comments to the Author

Reviewer #1: The submitted article addresses the prevalence of bovine tuberculosis which considered as one of the major problems affecting the dairy industry and attract the attention of the dairy producers and public health professionals. The study has been done on large number of herds (299) covering six different geographical areas in Central Ethiopia. The study used comparative intradermal tuberculin test for diagnosis of bovine tuberculosis. The authors provided the raw data and a copy of the survey but there are still some issues that are not clear and need to be addressed before considering publication. The manuscript requires major revision.

1- Title: “High level” Do you mean high prevalence?

2- Abstract:

- Line 23 “have been fragmented” what do you mean by previous studies have been fragmented?

- Line 24 “one-stage cluster” what was your cluster? Please, make it clear in the text.

- Line 25 “The survey, which was by far the largest in the area up to date” it is not clear if the authors mean that the survey was the largest in the number of question or the study was the largest in the area of coverage. Please, clarify.

- Line 26 &28” Tuberculin” and “Single Intradermal Cervical Comparative Tuberculin” you do not have to use capital letters.

- Line 35 “explore risk factors associated with reaction status” what does reaction status mean?

- Line 37 “ten times the odds of bTB detection” what does detection mean? The study did not isolate bTB.

- Line 43” within-herd prevalence between herds” is it within herd or between herds or both?

- Line 43-44 “findings that lays an important foundation for the future development of control strategies” Please clarify how?

-

3- Introduction:

- Line 73-74 “estimated the cost of bTB in Ethiopia for the urban livestock

production to have ranged from US$500,000–4.9 million between 2005 and 2011” is that the cost for the 6 years?

- Line 74 “These figures” which figures?

- Lines 81-83” As previous studies on bTB in this area had been fragmented and limited in coverage, there was a clear need to carry out a comprehensive survey to understand bTB in dairies in central Ethiopia” this is a conclusive statement but there is no mention to these studies and their limitations. It will be more useful for the reader to understand the knowledge gap that the paper will cover if the authors mention these studies and the limitation of each.

4- Material and methods:

- There is no mention of the program and the method used to create the map with the different prevalence’s (Fig 1).

- Statistical analysis:

o In the statistical analysis section, the authors mentioned that they used “Kruskal–Wallis test was used for comparison of variability in within herd bTB prevalence” It is not clear where in the results they mentioned that part. If they mean tables 2 and 3 A&B so where is the p-vales to show if there a significant difference or not?

o The formula for the GLMM is missing the intercept.

o In the formula it appears as the herd and area were crossed random effects. Are the area and herds as random effect crossed or nested? Please clarify.

o Why the authors did not use AIC for evaluation of the model goodness of fit?

o How the authors handled the animals with reading difference between 1-4 mm at the reactor and nonreactor scale for the GLMM?

5- Results:

I would recommend adding a section in the results to add the results of different sections of the survey because some terms are not clear in the manuscript here but has more description in the survey such as ventilation, housing, biosecurity measures.

For example, ventilation in the manuscript the classifications were very good, satisfactory and poor but there is no description for what each of these means while in the survey it is clearly described what each of these categories mean.

- P-value in the tables is mentioned beside the reference group but each level should have its P-value. For example, table 5 there are 3 levels for the herd size and 5 levels for age but only one P-value for the reference category. The model is comparing all the levels to the reference level so each level should have a P-value that could be different than the other levels.

- Data in table 5

Using of ( ] as exclusive and inclusive criteria is confusing I would recommend using the actual numbers for each group.

Age categories started at 0.1 but in the testing criteria the authors mentioned that they did not test animals less than 3 months of age. Please clarify.

6- Discussion

Comments are in the attached pdf copy.

7- Other comments are in the attached pdf copy.

8- Supplemented data:

Dataset (S1 Dataset Raw Data):

The authors provided an excel sheet for the dataset. There are some values in the data sheet that looks odd to me and it could be due to incorrect data entry such as:

Column (B2(mm) has values of 2278, 11262, 685.

Column (?A(mm) has value of 794.09

Column (?B(mm) has values of 2275.07, 11254.04

Column (?B-?A(mm) has values of 8.88E-16, 794.09

These measurements cannot be measured by tuberculin caliper. I would recommend the authors to revise their data entry as these values will affect the outcome of the study.

Reviewer #2: The study has relevance due to the fact to demonstrate that large herds is a major risk factor for bTB in the country and due to the fact that large herds are the main suppliers of heifers for small herds.

How is the bTB program in the country? It requires testing and culling? Movement control of the animals? The author should provide a brief of the Ethiopian program to the readers.

Line 75; add space after "[13]"

Line 105: add space after "[15]"

Line 233: correct "Prevalence of"

Line 316; the word becategorisedat must be correct "be categorized at"

Lines 401 to 406 – the comment may be included in discussion but is not a conclusion.

6. PLOS authors have the option to publish the peer review history of their article (what does this mean?). If published, this will include your full peer review and any attached files.

Reviewer #1: No

Reviewer #2: **Yes: **Paulo Alex Machado Carneiro

---

## [Author Response · Author response to Decision Letter 0]

8 May 2021

PONE-D-20-28942

High level of bovine tuberculosis in dairy herds of central Ethiopia: a call for intervention

PLOS ONE

Dear Dr. Almaw,

Thank you for submitting your manuscript to PLOS ONE. After careful consideration, we feel that it has merit but does not fully meet PLOS ONE’s publication criteria as it currently stands. Therefore, we invite you to submit a revised version of the manuscript that addresses the points raised during the review process.

Both expert reviewers have highlighted several concerns that preclude the acceptance of the manuscript as it stands. Please revise the manuscript accordingly making sure to address all reviewers' comments.

We look forward to receiving your revised manuscript.

Kind regards,

Angel Abuelo, DVM, MRes, MSc, PhD, DABVP (Dairy), DECBHM

Academic Editor

PLOS ONE

Journal Requirements:

Response: 

We have checked the revised manuscript against the PLOS ONE editorial requirements.

2. Please provide additional details regarding participant consent. In the ethics statement in the Methods and online submission information, please ensure that you have specified (1) whether consent was informed and (2) what type you obtained (for instance, written or verbal, and if verbal, how it was documented and witnessed). If the need for consent was waived by the ethics committee, please include this information.

Response:

This study was approved by AHRI-ALERT Ethics Review Committee (Project Reg.No PO46/14) and Ethiopia's National Research Ethics Review Committee (NRERC No. 3.10/800/07). Informed consent was obtained verbally from dairy farm owners who were briefed in the presence of a witness (local experts) on the tuberculin skin testing procedure; no known risks to the animal associated with this; their participation in study is voluntary; and that confidentiality on test result will be maintained. When agreed, the witness and the participant's full addresses including their mobile phone numbers were recorded for filing and in case contact with participant was needed.

"This research was financially supported by the Ethiopia Control of Bovine Tuberculosis

Strategies (ETHICOBOTS) project funded by the Biotechnology and Biological Sciences

Research Council, the Department for International Development, the Economic & Social

Research Council, the Medical Research Council, the Natural Environment Research Council

and the Defence Science &Technology Laboratory, under the Zoonoses and Emerging Livestock

Systems (ZELS) program, ref: BB/L018977/1.Stefan Berg was also funded by Defra, United

Kingdom, ref: TBSE3294"

 Response:

Funding-related text has been removed from the revised manuscript. The funding statement is included in the cover letter.

4. One of the noted authors is a group or consortium [ETHICOBOTS consortium]. In addition to naming the author group and listing the individual authors and affiliations within this group in the acknowledgments section of your manuscript, please also indicate clearly a lead author for this group along with a contact email address.

 Response:

The lead author: Professor James LN Wood; email: jlnw2@cam.ac.uk has been included

 Response:

All references to unpublished data have been removed in the revised manuscript, all raw data tables analysed in the study are provided as supplementary information.

6. We note that Figure 1 and S1Fig in your submission contain map images which may be copyrighted. All PLOS content is published under the Creative Commons Attribution License (CC BY 4.0), which means that the manuscript, images, and Supporting Information files will be freely available online, and any third party is permitted to access, download, copy, distribute, and use these materials in any way, even commercially, with proper attribution. For these reasons, we cannot publish previously copyrighted maps or satellite images created using proprietary data, such as Google software (Google Maps, Street View, and Earth). For more information, see our copyright guidelines: http://journals.plos.org/plosone/s/licenses-and-copyright.

6.1. You may seek permission from the original copyright holder of Figure 1 and S1Fig to publish the content specifically under the CC BY 4.0 license. 

6.2. If you are unable to obtain permission from the original copyright holder to publish these figures under the CC BY 4.0 license or if the copyright holder’s requirements are incompatible with the CC BY 4.0 license, please either i) remove the figure or ii) supply a replacement figure that complies with the CC BY 4.0 license. Please check copyright information on all replacement figures and update the figure caption with source information. If applicable, please specify in the figure caption text when a figure is similar but not identical to the original image and is therefore for illustrative purposes only.

 Response:

We developed a replacement map which is not copyrighted.

A free software program called Quantum Geographic Information System (QGIS) version 3.8 was used for compiling the maps. Administrative and road data were extracted and complied from publicly available information of Central Statistical Agency of Ethiopia (http://www.csa.gov.et/) and Ethiopian Roads Authority (www.era.gov.et). And a description at the bottom of the map saying: 'the administrative and political boundaries used here are for illustrative purpose and should not be considered authoritative' is included.

Reviewers' comments:

Reviewer's Responses to Questions

Comments to the Author

1. Is the manuscript technically sound, and do the data support the conclusions?

Reviewer #1: Yes

Reviewer #2: Yes

2. Has the statistical analysis been performed appropriately and rigorously? 

Reviewer #1: Yes

Reviewer #2: Yes

3. Have the authors made all data underlying the findings in their manuscript fully available?

Reviewer #1: Yes

Reviewer #2: Yes

4. Is the manuscript presented in an intelligible fashion and written in standard English?

Reviewer #1: Yes

Reviewer #2: Yes

5. Review Comments to the Author

Reviewer #1: The submitted article addresses the prevalence of bovine tuberculosis which considered as one of the major problems affecting the dairy industry and attract the attention of the dairy producers and public health professionals. The study has been done on large number of herds (299) covering six different geographical areas in Central Ethiopia. The study used comparative intradermal tuberculin test for diagnosis of bovine tuberculosis. The authors provided the raw data and a copy of the survey but there are still some issues that are not clear and need to be addressed before considering publication. The manuscript requires major revision.

1- Title: “High level” Do you mean high prevalence?

Response: 

Yes, the high level refers to prevalence. We have modified the title to make this point clearer: " The high and variable prevalence of bovine tuberculosis in dairy herds of central Ethiopia provides opportunities for targeted intervention." 

2-Abstract:

- Line 23 “have been fragmented” what do you mean by previous studies have been fragmented?

Response:

 Thank you for this comment. We have included additional information to make it clearer. By fragmented we meant that previous studies were small scale surveys (few farms/few areas) conducted over different time periods and study areas. So, there were concerns of representativeness. There was over/under representation of farms due to lack of either appropriate stratified sampling or standardization of the result accordingly. We have cited studies in the discussion part to support this. Taking the comment into account we have expanded the section to state: "Previous studies had been limited to few farms/study sites which lead to over/under representation of farms and failed to show the best estimate of bTB prevalence and the associated risk factors for initiating control intervention in central Ethiopia."

- Line 24 “one-stage cluster” what was your cluster? Please, make it clear in the text.

Response: 

It was in the text Line 112 "taking dairy herd as a cluster". Now we have included this phrase in the abstract too, to make it clear for abstract readers.

- Line 25 “The survey, which was by far the largest in the area up to date” it is not clear if the authors mean that the survey was the largest in the number of question or the study was the largest in the area of coverage. Please, clarify.

Response:

 By this we meant that compared to previous studies we have included a larger number of farms (#299) though this number is determined based on statistics as described in the manuscript. None of the previous surveys sampled more than 100 farms and from fewer study areas. We have also included a significant number of relevant questions as can be seen in the questionnaire format and most of them were used for analysis (# 16 risk factors).

We have modified to “The survey, which was to date by far the largest in the area in terms of number of dairy farms, study areas, and risk factors explored"

- Line 26 & 28” Tuberculin” and “Single Intradermal Cervical Comparative Tuberculin” you do not have to use capital letters.

Response: 

Corrected accordingly

- Line 35 “explore risk factors associated with reaction status” what does reaction status mean?

Response: 

By "reaction" to mean response to intradermal injection of bovine/avian tuberculin PPD. And reactors are those responding to the test (positive reaction), called reactors and those which did not respond called non-reactors. Common terms in bTB.

We changed to “explore risk factors associated with bTB status” to avoid ambiguity.

- Line 37 “ten times the odds of bTB detection” what does detection mean? The study did not isolate bTB.

Response: 

We have changed to: “...the odds of being a tuberculin reactor”. This will clarify the test was tuberculin skin test not isolation of the pathogen.

- Line 43” within-herd prevalence between herds” is it within herd or between herds or both?

Response: It is within herd prevalence.

- Line 43-44 “findings that lays an important foundation for the future development of control strategies” Please clarify how?

Response:

 Knowing the within herd prevalence variability, will help to know the extent of transmission in different herds which again is important which herds to target in bTB control strategy. In addition, establishing the best estimate of prevalence and identify potential risk factors is an input for designing a bTB control strategy. 

3-Introduction:

- Line 73-74 “estimated the cost of bTB in Ethiopia for the urban livestock

production to have ranged from US$500,000–4.9 million between 2005 and 2011” is that the cost for the 6 years?

Response: 

Not. the cost is simulated for each year from 2005 to 2011. For example US$500,000 is the estimated cost for the year 2005. We have clarified the point in the revised manuscript:

Tschopp and colleagues [13] estimated (simulated) the cost of bTB for the urban dairy production in central Ethiopia (Addis Ababa) to have ranged from US$500,000 - 4.9 million over a period of six years (2005-2011). 

-Line 74 “These figures” which figures?

Response: 

Addressed above in the prior response. 

- Lines 81-83” As previous studies on bTB in this area had been fragmented and limited in coverage, there was a clear need to carry out a comprehensive survey to understand bTB in dairies in central Ethiopia” this is a conclusive statement but there is no mention to these studies and their limitations. It will be more useful for the reader to understand the knowledge gap that the paper will cover if the authors mention these studies and the limitation of each.

Response: 

Addressed in the preceding section (Abstract section) and limitations of specific studies is mentioned in the discussion part of the revised manuscript Lines 319-325. But for the Introduction section we have modified to:

"Previous bTB prevalence studies in this part of Ethiopia were surveys of smaller scale (significantly fewer farms or fewer study areas) and conducted over different time periods and study areas, leading to concerns about representativeness. Accordingly, there is likely to have been over/under representation of dairy farms in past surveys due to lack of either appropriate stratified sampling or standardisation of the results [10]. A comprehensive review of bTB in Ethiopia by Sibhat et al. [12] showed limitations of previous prevalence studies, central Ethiopia included, including the scope of study objectives, methodology used, target population and geographic coverage.."

4- Material and methods:

- There is no mention of the program and the method used to create the map with the different prevalence’s (Fig 1).

Response: 

The paragraph below is included in the revised manuscript (with references):

"A free software program called Quantum Geographic Information System (QGIS) version 3.8 was used for compiling the maps. Administrative and road data were extracted and complied from publicly available information of Central Statistical Agency of Ethiopia and Ethiopian Roads Authority ."

- Statistical analysis:

o In the statistical analysis section, the authors mentioned that they used “Kruskal–Wallis test was used for comparison of variability in within herd bTB prevalence” It is not clear where in the results they mentioned that part. If they mean tables 2 and 3 A&B so where is the p-vales to show if there a significant difference or not?

Response:

 It was mentioned Line 271 (Kruskal–Wallis test: df=2,�2=33.295, p value < 0.001) of the previous manuscript. The result is presented in text form not in a table.

o The formula for the GLMM is missing the intercept.

Response: 

Thank you. We have revised and βo is now included.

 o In the formula it appears as the herd and area were crossed random effects. Are the area and herds as random effect crossed or nested? Please clarify.

Response: 

We treated them as crossed. This is because every herd (category) belongs to every study area and we consider this as crossed not nested (to one group or area only).

o Why the authors did not use AIC for evaluation of the model goodness of fit?

Response: 

Thank you, for catching this oversight! ROC analysis was used in the previous manuscript to assess the absolute goodness of fit (classification) ability of the estimated models. For model selection AIC, which only provides a relative measure of fit, should have been used for model selection. For the revised manuscript we have repeated the model selection procedure using AIC and clarified the use of AIC for model selection in the revised manuscript. 

In the process of carrying out these revisions we identified that the interaction effect between herd and breed was biasing estimates of other variables (skewing the estimate for the herd size variable) due to the small number of zebu cattle in the medium herd level category (# 37 VS 143 for small and 64 for large herds).

Dropping this interaction – which should have been screened out before based on the sparse distribution of breeds between herds – results in the global model having both the lowest AIC and highest Akaike weight. We thank the reviewer again for (indirectly) bringing attention to this error which has greatly improved the robustness of our results.

o How the authors handled the animals with reading difference between 1-4 mm at the reactor and nonreactor scale for the GLMM?

Response: 

They were treated as negatives for the binary data (outcome). This clarification is included in the revised manuscript.

5- Results:

I would recommend adding a section in the results to add the results of different sections of the survey because some terms are not clear in the manuscript here but has more description in the survey such as ventilation, housing, biosecurity measures.

For example, ventilation in the manuscript the classifications were very good, satisfactory and poor but there is no description for what each of these means while in the survey it is clearly described what each of these categories mean.

Response:

 We can. But we feel major results of the survey is presented in the result section. Additional information is refered in the text as "A full description of the measured risk factors is provided in S2 Table." This table if brought to the result section will increase the number of tables in the manuscript to 6.

- P-value in the tables is mentioned beside the reference group but each level should have its P-value. For example, table 5 there are 3 levels for the herd size and 5 levels for age but only one P-value for the reference category. The model is comparing all the levels to the reference level so each level should have a P-value that could be different than the other levels.

Response: 

P values are now included for all levels.

- Data in table 5 Using of ( ] as exclusive and inclusive criteria is confusing I would recommend using the actual numbers for each group.

Response: 

We agree and have edited the table as suggested replacing brackets and parenthesis with > or < expression so that it can be easily understood. For example for the age category: 

 (4,20] >4 to ≤20 

 (20,37] >20 to ≤37 

 (37,168] >37 to ≤168 

 Age categories started at 0.1 but in the testing criteria the authors mentioned that they did not test animals less than 3 months of age. Please clarify.

Response: 

Thank you for catching this error. We have corrected the manuscript with respect to excluding animals less than 6 weeks of age.

6-Discussion

Comments are in the attached pdf copy.

Response: Edited as per the comments

7- Other comments are in the attached pdf copy.

Response: Edited as per the comments

8- Supplemented data:

Dataset (S1 Dataset Raw Data):

The authors provided an excel sheet for the dataset. There are some values in the data sheet that looks odd to me and it could be due to incorrect data entry such as:

Column (B2 (mm) has values of 2278, 11262, 685.

Column (?A (mm) has value of 794.09

Column (?B (mm) has values of 2275.07, 11254.04

Column (?B-?A (mm) has values of 8.88E-16, 794.09

These measurements cannot be measured by tuberculin caliper. I would recommend the authors to revise their data entry as these values will affect the outcome of the study.

Response: 

Cross-checked with the hard copy and all corrected (shown in brackets below). We have checked them all and these were the only changes (# 4 animals). It was a mistake when entering the decimal point. We have not used the actual measurements for analysis. We used the dichotmous (FALSE coded as "O" for negatives or TRUE coded as 1 for positives) and we found no change in result and not affected. We resubmitted the edited version shown here in brackets.

Column (B2 (mm) has values of 2278 (22.78), 11262 (12.62), 685 (68.5).

Column (?A (mm) has value of 794.09: Due to Column (A2 (mm) 802 (8.02)

Column (?B (mm) has values of 2275.07, 11254.04: associated with value of Column (B2 (mm)

Column (?B-?A (mm) has values of 8.88E-16, 794.09: associated with value of Column (A2 (mm)

Reviewer #2: The study has relevance due to the fact to demonstrate that large herds is a major risk factor for bTB in the country and due to the fact that large herds are the main suppliers of heifers for small herds.

How is the bTB program in the country? It requires testing and culling? Movement control of the animals? The author should provide a brief of the Ethiopian program to the readers.

Response: 

 A brief comment has been included in the discussion.

Line 75; add space after "[13]"

Response: 

 corrected.

Line 105: add space after "[15]"

Response: corrected.

Line 233: correct "Prevalence of"

Response: 

corrected.

Line 316; the word becategorisedat must be correct "be categorized at"

Response: 

corrected.

Lines 401 to 406 – the comment may be included in discussion but is not a conclusion.

Response: 

 Accepted. This part is removed from the conclusion and moved to the discussion part in the revised manuscript.________________________________________

6. PLOS authors have the option to publish the peer review history of their article (what does this mean?). If published, this will include your full peer review and any attached files.

Do you want your identity to be public for this peer review? For information about this choice, including consent withdrawal, please see our Privacy Policy.

Reviewer #1: No

Reviewer #2: Yes: Paulo Alex Machado Carneiro

• 

PONE-D-20-28942_reviewer.pdf

2.8MB

---

## [Decision Letter · Decision Letter 1]

14 Jun 2021

PONE-D-20-28942R1

The high and variable prevalence of bovine tuberculosis in dairy herds of central Ethiopia provides opportunities for targeted intervention

PLOS ONE

Dear Dr. Almaw,

Thank you for submitting your manuscript to PLOS ONE. After careful consideration, we feel that it has merit but does not fully meet PLOS ONE’s publication criteria as it currently stands. Therefore, we invite you to submit a revised version of the manuscript that addresses the points raised during the review process.

Please make sure that you address all the pending issues highlighted by reviewer #1 in the attached document.

We look forward to receiving your revised manuscript.

Kind regards,

Angel Abuelo, DVM, MRes, MSc, PhD, DABVP (Dairy), DECBHM

Academic Editor

PLOS ONE

Journal Requirements:

Reviewers' comments:

Reviewer's Responses to Questions

**Comments to the Author**

1. If the authors have adequately addressed your comments raised in a previous round of review and you feel that this manuscript is now acceptable for publication, you may indicate that here to bypass the “Comments to the Author” section, enter your conflict of interest statement in the “Confidential to Editor” section, and submit your "Accept" recommendation.

Reviewer #1: (No Response)

Reviewer #2: All comments have been addressed

2. Is the manuscript technically sound, and do the data support the conclusions?

Reviewer #1: Yes

Reviewer #2: Yes

3. Has the statistical analysis been performed appropriately and rigorously? 

Reviewer #1: Yes

Reviewer #2: Yes

4. Have the authors made all data underlying the findings in their manuscript fully available?

Reviewer #1: Yes

Reviewer #2: Yes

5. Is the manuscript presented in an intelligible fashion and written in standard English?

Reviewer #1: Yes

Reviewer #2: Yes

6. Review Comments to the Author

Reviewer #1: There is a significantly improvement of the manuscript after addressing the previous comments. There are few comments in the attached pdf document that has not been addressed from the previous revision, please address them.

I would recommend the title to be modified to "The variable prevalence of bovine tuberculosis among dairy herds in Central Ethiopia provides opportunities for targeted intervention" as variable as aword includes both high and low values.

Reviewer #2: (No Response)

7. PLOS authors have the option to publish the peer review history of their article (what does this mean?). If published, this will include your full peer review and any attached files.

Reviewer #1: No

Reviewer #2: **Yes: **Paulo Alex Machado Carneiro

---

## [Author Response · Author response to Decision Letter 1]

15 Jun 2021

PONE-D-20-28942R1

The high and variable prevalence of bovine tuberculosis in dairy herds of central Ethiopia provides opportunities for targeted intervention

PLOS ONE

Dear Dr. Almaw,

Thank you for submitting your manuscript to PLOS ONE. After careful consideration, we feel that it has merit but does not fully meet PLOS ONE’s publication criteria as it currently stands. Therefore, we invite you to submit a revised version of the manuscript that addresses the points raised during the review process.

Please make sure that you address all the pending issues highlighted by reviewer #1 in the attached document.

Response: All comments highlighted in the PDF file by reviewer # 1 are addressed point by point in the PDF file itself and also in the Revised Clean Manuscript and Revised Manuscript with Track Changes. The PDF file with responses to comments is now uploaded separately as "Response to Reviewers" file. Also we have sent it via email to avoid changes (if any) in the highlighted responses during PDF creation in the online submission system. 

We look forward to receiving your revised manuscript.

Kind regards,

Angel Abuelo, DVM, MRes, MSc, PhD, DABVP (Dairy), DECBHM

Academic Editor

PLOS ONE

Journal Requirements:

Response: We have checked the reference list and it is complete and correct. We have not cited papers that have been retracted.

Reviewers' comments:

Reviewer's Responses to Questions

Comments to the Author

1. If the authors have adequately addressed your comments raised in a previous round of review and you feel that this manuscript is now acceptable for publication, you may indicate that here to bypass the “Comments to the Author” section, enter your conflict of interest statement in the “Confidential to Editor” section, and submit your "Accept" recommendation.

Reviewer #1: (No Response)

Reviewer #2: All comments have been addressed

2. Is the manuscript technically sound, and do the data support the conclusions?

Reviewer #1: Yes

Reviewer #2: Yes

3. Has the statistical analysis been performed appropriately and rigorously? 

Reviewer #1: Yes

Reviewer #2: Yes

4. Have the authors made all data underlying the findings in their manuscript fully available?

Reviewer #1: Yes

Reviewer #2: Yes

5. Is the manuscript presented in an intelligible fashion and written in standard English?

Reviewer #1: Yes

Reviewer #2: Yes

6. Review Comments to the Author

Reviewer #1: There is a significantly improvement of the manuscript after addressing the previous comments. There are few comments in the attached pdf document that has not been addressed from the previous revision, please address them.

Response: Addressed. This time we addressed point by point in the PDF file itself. In the previous revision we addressed these comments only in the Clean Manuscript and Revised Manuscript with Track Changes and we failed to link these responses to the PDF file comments. Now we have included Line numbers, Tables and Pages in the point by point responses in the PDF file for tracking responses easily. The PDF file with responses to comments is now uploaded separately as "Response to Reviewers" file. Also we have sent PDF file with point by point responses via email to the Academic Editor.

I would recommend the title to be modified to "The variable prevalence of bovine tuberculosis among dairy herds in Central Ethiopia provides opportunities for targeted intervention" as variable as a word includes both high and low values.

Response: Thank you! Accepted and title modified as recommended.

Reviewer #2: (No Response)

7. PLOS authors have the option to publish the peer review history of their article (what does this mean?). If published, this will include your full peer review and any attached files.

Do you want your identity to be public for this peer review? For information about this choice, including consent withdrawal, please see our Privacy Policy.

Reviewer #1: No

Reviewer #2: Yes: Paulo Alex Machado Carneiro

---

## [Editor Report · Decision Letter 2]

21 Jun 2021

The variable prevalence of bovine tuberculosis among dairy herds in Central Ethiopia provides opportunities for targeted intervention

PONE-D-20-28942R2

Dear Dr. Almaw,

We’re pleased to inform you that your manuscript has been judged scientifically suitable for publication and will be formally accepted for publication once it meets all outstanding technical requirements.

Kind regards,

Angel Abuelo, DVM, MRes, MSc, PhD, DABVP (Dairy), DECBHM

Academic Editor

PLOS ONE
---

## [Editor Report · Acceptance letter]

24 Jun 2021

PONE-D-20-28942R2 

The variable prevalence of bovine tuberculosis among dairy herds in Central Ethiopia provides opportunities for targeted intervention 

Dear Dr. Almaw:

I'm pleased to inform you that your manuscript has been deemed suitable for publication in PLOS ONE. Congratulations! Your manuscript is now with our production department. 

Kind regards, 

on behalf of

Dr. Angel Abuelo 

Academic Editor

PLOS ONE